# Bio-Inspired Aramid Fibers@silica Binary Synergistic Aerogels with High Thermal Insulation and Fire-Retardant Performance

**DOI:** 10.3390/polym15010141

**Published:** 2022-12-28

**Authors:** Jinman Zhou, Xianyuan Liu, Xiaojiang He, Haoxin Wang, Dongli Ma, Xianyong Lu

**Affiliations:** 1Key Laboratory of Bio-Inspired Smart Interfacial Science and Technology of Ministry of Education, School of Chemistry, Beihang University, Beijing 100191, China; 2Beijing Huateng Rubber Plastic & Latex Products Co., Ltd., Beijing 101100, China

**Keywords:** aramid nanofibers, bionic structure, silica, heat resistance, flame retardancy

## Abstract

Flame-retardant, thermal insulation, mechanically robust, and comprehensive protection against extreme environmental threats aerogels are highly desirable for protective equipment. Herein, inspired by the core (organic)-shell (inorganic) structure of lobster antenna, fire-retardant and mechanically robust aramid fibers@silica nanocomposite aerogels with core-shell structures are fabricated via the sol-gel-film transformation and chemical vapor deposition process. The thickness of silica coating can be well-defined and controlled by the CVD time. Aramid fibers@silica nanocomposite aerogels show high heat resistance (530 °C), low thermal conductivity of 0.030 W·m^−1^·K^−1^, high tensile strength of 7.5 MPa and good flexibility. More importantly, aramid fibers@silica aerogels have high flame retardancy with limiting oxygen index 36.5. In addition, this material fabricated by the simple preparation process is believed to have potential application value in the field of aerospace or high-temperature thermal protection.

## 1. Introduction

Aerospace products often use multilayer structures to resist extreme temperature differences and solar radiation in space. Although multilayer structures can protect against multiple threats, a single multifunctional material can reduce the load-bearing weight and manufacturing complexity [1,2]. Traditional thermal insulation and mechanical energy dissipation mechanisms compete with each other, so an effective composite of material structures is needed to break this competition mechanism [2]. In the space environment, many personal thermal protection systems urgently need high-performance insulation materials. At present, various advanced insulation materials have been developed, including mineral wool, glass fiber, polymer-based foams, and aerogels [3,4,5,6]. Among them, aerogels are considered one of the ideal candidate materials due to their high porosity, large specific surface area, low apparent density, and low thermal conductivity [7,8,9,10]. Silica aerogels [11,12,13], carbon aerogels [14,15], metal oxide aerogels [16,17], and organic polymers aerogels [18,19,20] have been well-explored in recently years. Among the classes of aerogels, silica aerogel exhibits the most remarkable physical properties. As well known, the frameworks of silica aerogels with open porous architectures consist of interconnected silica nanoparticles [6,12,21]. The weak interfacial interactions between nanoparticles and silica aerogels usually suffer from the fragile property, low mechanical strength, and poor flexibility, thus severely restricting their broad applications [9,22]. However, traditional organic aerogel is difficult to apply in high-temperature thermal protection systems due to its low mechanical strength, flammability, and high-temperature resistance [23]. Therefore, the development of porous nanomaterials with low thermal conductivity, excellent mechanical properties, high flexibility, and good heat resistance has been a problem continuously explored by scientists [21,23].

The ultra-high strength and fatigue resistance make aramid fibers one of the best choices to meet the demanding requirements of many cutting-edge fields, including aerospace, electronics, tank, and bulletproof products [2,24,25]. Aramid fibers have highly symmetric and extended molecular chains that interact through a network of the firm and highly aligned hydrogen bonds, exhibiting high mechanical strength and chemical inertness. In 2011, Kotov [26] and his colleagues successfully prepared an aramid nanofiber solution by deprotonating poly-p-phenylene terephthamide (PPTA) fibers. Due to the physical entanglement and strong hydrogen bond interaction between nanofibers, ANFs have excellent mechanical strength and toughness [27]. ANFs began to be used to prepare multifunctional hybrid materials, including supercapacitors [28,29,30], sensors [31,32], thermal managers [8,18,33,34], infrared stealth materials [20,35], insulation materials [36,37], and electromagnetic interference shielding materials [38]. It is worth noting that aramid fiber, as one of the most heat-resistant organic materials, is widely used in heat insulation materials and engineering materials [19,39], but is easily ignitable in the absence of fire barriers [6,35,40]. Polymer aerogels with flame retardants could sustain more time; however, flame retardants are environmentally unfriendly [41,42]. Therefore, coating inorganic materials on the surface of organic insulation materials is an efficient strategy for preparation materials with low thermal conductivity and a high fire resistance [23,43,44]. Recently, carbon nanotubes, silica, metal oxides et al. as inorganic fillers have been reported to prepare organic–inorganic composites with flame-retardant properties [23]. The phase compatibility and the distribution of organic–inorganic materials are challenging for preparation of their composites [22,31,45].

The antennae of lobster have the core (organic)-shell (inorganic) structure and flexible property that inspired us to design a strategy for novel polymer-silica hybrid aerogels with well-defined thermal insulation and fire-retardant performance. In our work, we developed a bioinspired-strategy to coat aramid nanofibers aerogel with a silica shell by chemical vapor deposition (CVD) of tetraethyl orthosilicate (TEOS) catalyzed by ammonia. Silica shell is dense and formed on aramid nanofibers by this method. Bioinspired aramid fibers@silica binary synergistic aerogels perform heat resistance (530 °C), low thermal conductivity (0.030 W·m^−1^·K^−1^), high fracture strength (7.5 MPa, 29.3% higher than that of pure ANFs aerogel), and are flame-retardant.

## 2. Materials and Methods

### 2.1. Materials

Kevlar 29 fibers were purchased from DuPont Co., Ltd. (Wilmington, DE, USA). Chemicals such as dimethyl sulfoxide (DMSO), tetraethyl orthosilicate (TEOS), potassium hydroxide (KOH), ethanol, and ammonia were purchased from Shanghai Aladdin Biochemical Technology Co., Ltd., China. All analytical chemical reagents were used as received without further purification. Deionized water was self-made.

### 2.2. Preparation of Aramid Nanofibers Aerogel

The aramid nanofiber solution was obtained according to a published work [26]. Firstly, 10 g bulk Kevlar 29 and 15 g potassium hydroxide (KOH) was added into 1 L dimethyl sulfoxide (DMSO). The dark red aramid nanofiber solution (10 g/L) was obtained after being magnetically stirred one week at room temperature.

The ANF/DMSO dispersion (1.58 g/L) was obtained by dilution of above-mentioned aramid nanofiber solution (10 g/L) with DMSO. The deionized water was added into ANF/DMSO dispersion (1.58 g/L) with volume ratio 1:0.95 for mixed protonated aramid nanofiber gel. The aramid hydrogel was prepared by the vacuum-filtration assisted layer-by-layer strategy with PTFE membrane (Pore size: 200 nm). The aramid nanofiber aerogel was prepared by CO_2_ supercritical drying.

### 2.3. Fabrication of Aramid Fibers@silica Aerogel

The ANFs aerogels were placed in a desiccator with two open glass vessels containing about 10 mL of tetraethyl orthosilicate (TEOS) and aqueous ammonia solution, respectively. The chemical vapor deposition was performed at the vacuum of 20 kPa and room temperature (25 °C). In order to control the deposition thickness of silica on the ANFs, the different reaction time were performed, respectively.

### 2.4. Characterizations

The Fourier transform infrared (FT-IR) spectroscopy was taken by PerkinElmer Frontier FT-IR infrared spectrometer (PerkinElmer Inc., Waltham, MA, USA) under ambient conditions. The scanning electron microscopic (SEM) images were performed by a Hitachi SU8010 cold-field emission scanning electron microscope (Hitachi, Japan) at an acceleration voltage of 5 kV.

Tensile stress–strain curves were recorded at a loading rate of 5 mm/min (Instron 5565A universal testing machine, INSTRON, Norwood, MA, USA) under 40% humidity and temperature of 25 °C. The sample length and width for all the samples were 20, 5 mm, respectively. The thickness of samples was measured by a film thickness meter (BK-3281, Shanghai Jihui Industrial Co., Ltd., Shanghai, China)To make the measurements as accurate as possible, we calculated the average thickness and standard deviation of the film by measuring the thickness at six different locations on the sample. The average mechanical properties and corresponding standard deviations for each sample type were obtained based on five measurement results.

Thermogravimetric analysis (TGA) was carried out on a Perkin-Elmer TGA 5500 (PerkinElmer Inc., USA). The samples were analyzed in a temperature range of 20–700 °C with a temperature increasing velocity of 5 °C/min under protection of nitrogen. The Oxygen index measuring instrument FTT0077 (Fire Testing Technology Limited, East Grinstead, UK) was used to test the limiting oxygen index of as-prepared samples by GB/T2406.2 standard. The sample size was 50 mm × 15 mm × 0.08 mm. Hot Disk TPS 2500 S thermal conductivity meter from Hot Disk Company in Sweden was used for the thermal conductivity test. The thermal imager (H21Pro, Hikvision Co., Hangzhou, China) was used to obtain infrared thermal images.

## 3. Results and Discussion

Figure 1 shows the fabrication of bioinspired aramid fibers@silica aerogel. The aramid fibers’ dark red solution was obtained by destroying the hydrogen bonding interaction between the backbone of PPTA molecules in the KOH/DMSO [26]. The aramid fibers have been reported as building blocks for high performance composited materials by layer-by-layer bioinspired fabrication strategy [24,27,46]. Among them, the vacuum-filtration assisted layer-by-layer strategy has been proved a powerful method for high mechanical polymer composited films [47]. In order to increase the critical size of polymer blocks, equal volume water was added into ANFs solution to promote the re-protonation process. The as-prepared ANFs mixture served as new building blocks for fabrication of multilayered ANFs hydrogel films after exchanging DMSO and KOH by water three times. The layered ANFs aerogel was prepared by CO_2_ supercritical drying.

### 3.1. Morphology and Composition Characterization of Aramid Fibers@silica Aerogel

The SEM images of as-prepared ANFs aerogel shows in Figure 2a. The ANFs had a diameter of 26.48 ± 0.68 nm and length of 11.75 ± 1.34 µm from statistical data of SEM images. The ANFs aerogel had well-defined three-dimensional structures, which benefits for its high thermal insulation property. According to the literature on aramid fiber, aramid fiber has good heat resistance (the initial decomposition temperature is 400~430 °C) due to the benzene ring structure and intermolecular hydrogen bond [33,48]. However, because of the time–temperature superposition principle of macromolecule motion, it is difficult for ANFs to maintain stable mechanical properties for a long time at high temperature (above 300 °C) [49]. Sometimes it is necessary to add thermostable fillers to ANFs for heat-resistance applications [22,39]. Inspired by the core (organic)-shell (inorganic) structure of the antennae of lobster, the as-prepared ANF aerogel were modified with SiO_2_ by a chemical vapor deposition (CVD). The ANFs have amide groups in the skeletons of poly-p-phenyleneterephthamide. In the process of CVD, the TEOS and ammonia molecules could easily approach the ANFs. Furthermore, SiO_2_ coating could be well-distributed formed on the surface of ANFs. Figure 2b,c shows the typical aramid fibers@silica aerogels obtained by CVD 8 h and 24 h, respectively. The statistical results indicate that the diameter of aramid fibers coated silica obtained by CVD 8 h and 24 h were 71.17 ± 1.16 nm and 114.95 ± 1.24 nm respectively. The statistical results in Appendix A indicate that the thickness of silica coating and the density of aramid fibers@silica aerogels could be controlled by the time of CVD. In addition to that, the silica has also physically fixed the aramid nanofibers that benefit from the better mechanical properties.

The FTIR spectra of the ANFs and the ANFs/TEOS-8 (in Figure 3, the aramid fibers@silica aerogel (CVD 8 h) were well-performed. Compared with Kevlar fiber, the transmittance and sharpness of the IR peaks of ANFs were decreased due to the broader distribution of bond lengths and surface states of the fibers [26]. For the ANFs, the characteristic peaks that were located at 3326.1 cm^−1^ and 722.4 cm^−1^ were ascribed to the stretching vibration and out-of-plane bending vibration of the N-H bond of the aramid nanofibers amide group. The characteristic peaks of the C=O stretching vibration of aramid nanofibers amide groups appeared at 1652.4 cm^−1^ in Figure 3. The peaks that were located at 1514.7 cm^−1^ and 1316.0 cm^−1^ were ascribed to the stretching vibration of C=C groups of the benzene ring and stretching vibration of Ph-N. For the ANFs/TEOS-8 aerogels, broad peaks located at 1068.6 cm^−1^ were observed, which were due to the formation of a large number of Si-O-Si and Si-O groups of aramid fibers@silica aerogels [45,50]. As shown in the FTIR spectra of the ANFs/TEOS-8, a wide weak peak with a certain degree of redshift appeared at the position where the characteristic peak of stretching vibration of N-H ought to occur, which indicates that the intermolecular hydrogen bond was formed between the Si-O group of silica and the N-H on the surface of the aramid nanofibers amide group.

### 3.2. Heat Resistance of Aramid Fibers@silica Aerogel

Thermogravimetric analysis (TGA) and Differential scanning calorimetry (DSC) were performed to study the thermal stability of ANFs aerogel and aramid fibers@silica aerogels. According to the thermogravimetric curves in Figure 4a, pure ANFs aerogel showed high thermal stability before 410 °C without significant weight loss. Compared with other fibers, aramid fiber has high thermal stability due to the intrinsic stability of repeated benzene units in the molecular backbone and the formation of stable hydrogen bonds between β-folded amide groups by adjacent chains [27]. For the aramid fibers@silica aerogels, the weight loss before 200 °C was attributed to the evaporation of water in the 3D porous fiber. The maximum mass loss rate was found at around 520 °C due to the thermal cracking of aramid fiber. The loading of silica could increase the decomposition temperature of composite fiber. As shown in Appendix A, compared with ANFs aerogel, the decomposition temperature of aramid fibers@silica aerogels prepared by CVD 4, 8, 12 and 24 h increased by 6.92%, 11.56%, 4.84%, and 5.35%, respectively. Statistics showed that the aramid fibers@silica aerogels still have 60–85% residual mass at 700 °C due to the silica superficial coating hindrance to the carbonization of the inner aramid nanofiber. It is worth noting that ANFs/TEOS-8 not only has the highest decomposition temperature of 530.8 °C and the highest initial decomposition temperature of 467.4 °C in Figure 4b, but also has a higher mass retention rate at higher temperatures. It might be related to the interaction between silica and 3D aramid nanofibers network. Excessive silica thickens the fiber diameter and reduces the porosity, resulting in a reduction in the mutual binding ability of the shell-core.

### 3.3. Mechanical Properties of Aramid Fibers@silica Aerogel

The mechanical properties of ANFs and ANFs/TEOS aerogels is shown in Figure 5. The average tensile strength, elongation at break and Young’s modulus of ANFs aerogel films were 5.8 ± 0.3 MPa, 11.7 ± 0.6%, and 49.57 ± 4.21 MPa, respectively. Silica aerogels usually suffer from fragile property, low mechanical strength, and poor flexibility due to the weak interfacial interactions between nanoparticles [6,12,21]. Surprisingly, the fracture strength of ANFs/TEOS in our work was greater than 4 MPa (Figure 5b), and the Young’s modulus was also improved, which was attributed to the performance directional laminar of aramid nanofiber structure formed by vacuum filtration. ANFs, ANFs/TEOS-4, and ANFs/TEOS-8 showed no obvious damage after folding and unfolding in Figure 5c, indicating their well-performed mechanical solidity and flexibility. The aramid fibers@silica aerogels obtained by longer CVD time broke after folding due to excessive silica coating. The average tensile strength, elongation at break, and Young’s modulus of ANFs/TEOS-8 aerogel were 7.5 ± 0.5 MPa, 9.9 ± 0.2%, and 76.14 ± 5.70 MPa, respectively. Compared with ANFs aerogel, the tensile strength and Young’s modulus were significantly increased by 29.3% and 53.6%, respectively. The coating silica on the surface of ANFs can increase the damping between fiber slip under dynamic stretching and enhance its tensile strength on the basis of maintaining the mechanical properties of the original layered nanofibers. Moreover, the presence of shell-core structure and fixation of the nanofiber joint can hinder crack propagation to some extent, and Young’s modulus is further promoted.

We prepared non-putamen aerogel samples by chemical liquid deposition, and silica particles with large size were densely stacked on the surface of ANFs aerogels (Appendix A). The mechanical properties of ANFs/TEOS aerogel with non-putamen structure (ANFs/TEOS-non), ANFs/TEOS aerogel with shell-core structure (ANFs/TEOS) and ANFs are shown in Appendix A. The silica coating in the two structures possibly affects the mechanical properties through two opposite mechanisms [22]. In the shell-core structure, the silica particles can not only act as the crosslinking agent, but also improve the sliding resistance between the fibers, so as to strengthen the interaction between the aramid nanofibers. In the non-putamen structure, the stress concentration induced by silica particles leads to the poor mechanical properties of composite aerogel.

### 3.4. Thermal Insulation and Flame Retardancy of Aramid Fibers@silica Aerogel

We used Hot Disk TPS 2500 S thermal conductivity meter (Hot Disk Company, Göteborg, Sweden) to measure the interlayer thermal conductivity of the fully dried sample, and calculated the average value of the two tests. As the statistical data show in Figure 6f, the thermal conductivity of ANFs aerogel in our work was 0.04022 W·m^−1^·K^−1^. The skeleton of ANFs contains a symmetric benzene ring structure, which helps to form an oriented and ordered layer structure during vacuum filtration. Heat is transported along the layer of the backbone, forming a continuous and stable heat conduction pathway. As the time of CVD increased, the thermal conductivity of the aerogel first decreased and then increased, reaching the minimum at the CVD 8 h. The thermal conductivity of ANFs/TEOS-8 was 0.03017 W·m^−1^·K^−1^, which was 25.0% lower than that of ANFs aerogel due to the fluffy silica particles blocking heat transfer. However, with the CVD time increasing, the diameter of the shell-core nanofibers became thicker, which reduced the thermal conductivity due to low porosity. Moreover, the silica coating on the fiber surface was gradually dense, the heat transfer between the layers was more efficient, and the thermal conductivity of the composite film was well-defined and improved.

To evaluate the thermal management capability of aerogels, an infrared thermal imaging system was used to record the temperature variation of the aerogel surface on a uniformly heated plate. Heating plate as a plane to heat one side of the aerogels, and the temperature change on the other side can reflect the thermal management ability of the film. As shown in Figure 6b–e, the center temperature of ANFs/TEOS-8 was lower than that of other aerogels at each time. The temperature management ability of different aerogel was in line with the change in their thermal conductivity.

The application of aerogel film as a thermal insulator requires not only good mechanical properties and heat resistance, but also excellent flame retardancy. The oxygen index of ANFs/TEOS-8 and pure ANFs was measured with a sample strip of 50 mm × 15 mm × 0.082 mm. Oxygen index greater than 28 belongs to flame-retardant materials. The oxygen index of ANFs/TEOS-8 composite membrane reached 36.45, which was 39.6% higher than that of pure ANFs (oxygen index 26.10). The greatly improved flame retardant was attributed to the thermal protection of the inner aramid nanofiber by dense silica coating. Silica as flame-retardant materials, its Si-O structure will transform into a continuous, oxidation-resistant, insulating network of silica ash covering the surface. ANFs/TEOS-8 aerogel has excellent self-extinguishing ability in Figure 7b. Therefore, silica coating is an effect strategy for ANFs aerogel film with well-performing heat resistance, mechanical property and heat insulation.

### 3.5. Comprehensive Performance Analysis of Aramid Fibers@silica Aerogel

In addition, physical properties of the ANFs/TEOS aerogel composite film including tensile strength, elongation at break, Young’s modulus, thermal conductivity, and heat-resistance temperature are compared in Figure 7c. The radar chart shows the good performance in flame retardancy and mechanical property of ANFs/TEOS-8 composite aerogel, which has the potential for applications in harsh flame environments, such as thermal insulators, shock absorbers, and flexible electronic equipment for fire protection [8,23].

The CVD method for in situ silica coating on ANFs aerogels is a powerful strategy for high performance ANFs/TEOS aerogels. The silica has been continuously coated on the surface of ANFs. It is essential for studying the binary synergistic effect between silica and ANFs in the process of CVD. Firstly, ammonia and water molecules were volatilized in a vacuum device. Water molecules make ammonia easily adsorbed on the surface of three-dimensional porous aramid nanofibers. TEOS gas met with ammonia for silica coating on the surface of aramid fiber. The statistical results indicate the thickness of silica coating, physical properties, mechanical properties, and thermal properties of aramid fibers@silica aerogels could be controlled by the time of CVD (Table 1). In addition to that, ANFs/TEOS aerogels inherit the high tensile strength due to silica nodes connected aramid nanofibers (Figure 8b) [51]. As shown in Figure 8b, the regular tetrahedral molecular structure results in very stable chemical properties of silica and excellent corrosion and heat resistance. The shell-core structure provided a continuous thermal and oxidation protective layer for the inner core aramid fiber, and the fluffy porous fiber structure could effectively block heat conduction. As a typical sample, the ANFs/TEOS-8 was heated to 800 °C at a rate of 3 °C/min, held for 4 h, and carbonized in the argon atmosphere. After carbonization, the aerogel maintained a three-dimensional network structure and the diameter of fiber became thinner (Appendix A) and showed an electrical conductivity of 0.37 S·cm^−1^ due to the interior carbon fiber. Therefore, this bioinspired novel ANFs/TEOS aerogels have high flexibility, mechanical strength, thermal stability, heat insulation, and are excellent flame-retardants.

## 4. Conclusions

In summary, a novel bioinspired ANFs/TEOS aerogels film was obtained by CVD of TEOS catalyzed by ammonia. The ANFs/TEOS aerogels film exhibited mechanical robustness, low thermal conductivity, high-temperature resistance, and excellent flame retardance at a specific composite ratio, which is attributed to the structural synergy between the ANFs frame and the silica surface layer. This ANFs/TEOS is a promising high-temperature thermal protection material due to its excellent comprehensive properties and ease of production.

## Figures and Tables

**Figure 1 polymers-15-00141-f001:**
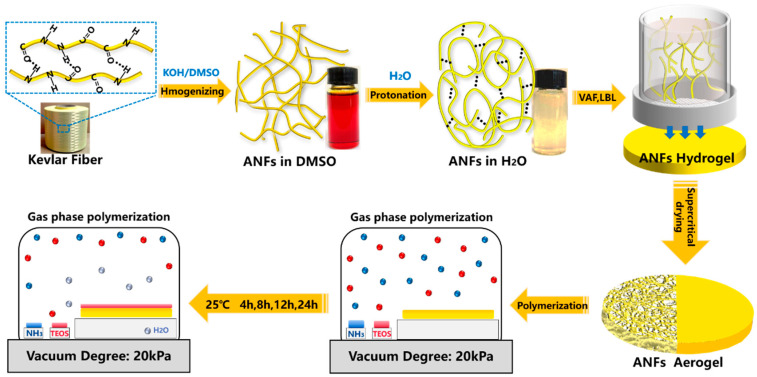
Schematic diagram of fabrication process of aramid fibers@silica aerogel.

**Figure 2 polymers-15-00141-f002:**
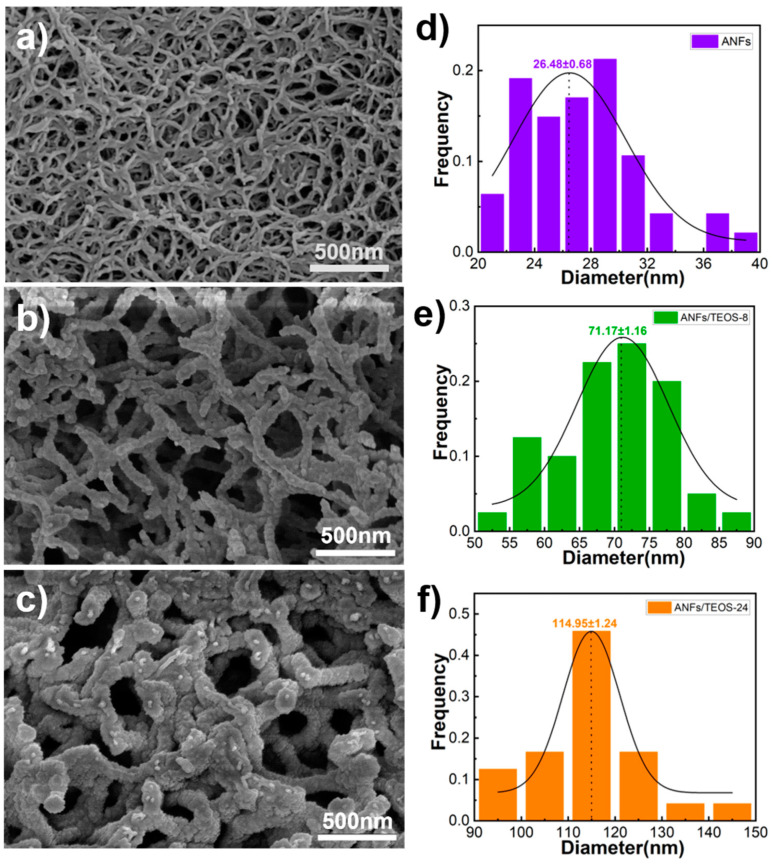
SEM images of ANFs aerogels (**a**), aramid fibers@silica aerogel prepared by CVD 8 h, (**b**) and 24 h (**c**), together with the histograms of the corresponding nanofibers (**d**–**f**), respectively.

**Figure 3 polymers-15-00141-f003:**
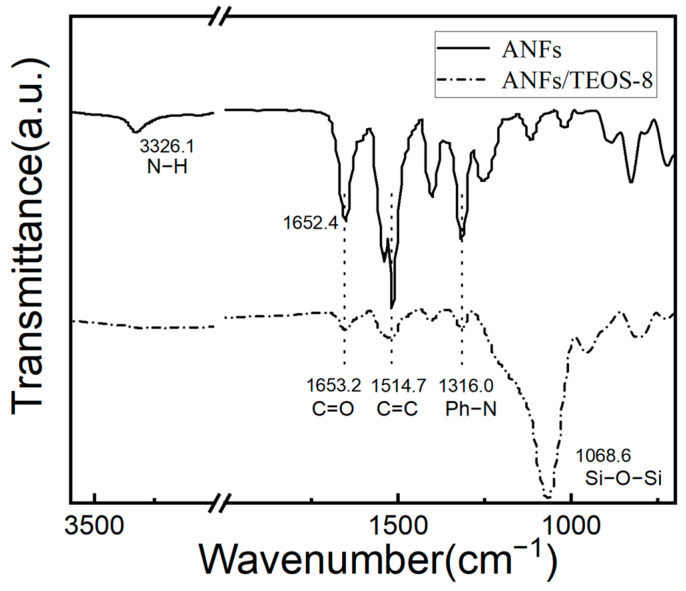
FTIR spectra of the ANFs aerogel and the aramid fibers@silica aerogel obtained by CVD 8 h with main peaks assignment.

**Figure 4 polymers-15-00141-f004:**
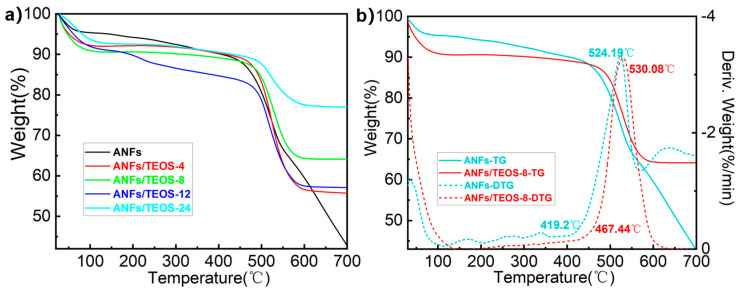
(**a**) The TG curves of ANFs aerogel and aramid fibers@silica aerogels prepared by different CVD times from 25 °C to 700 °C at oxygen atmosphere. (**b**) The comparison of TG and DTG curves between ANFs aerogel and aramid fibers@silica aerogel prepared by CVD 8 h.

**Figure 5 polymers-15-00141-f005:**
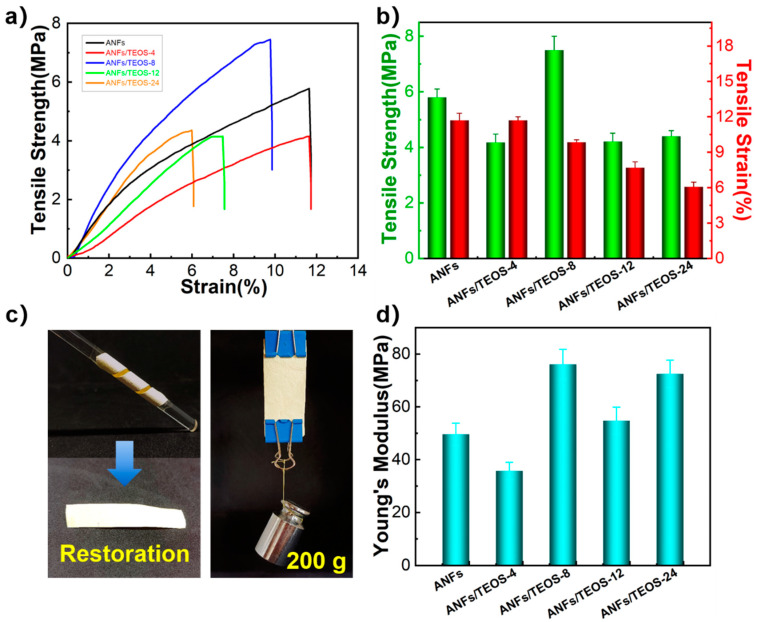
Mechanical performance of aerogels. (**a**) Stress–strain curves, (**b**) Tensile strength and elongation at break, (**c**) Flexibility and mechanical strength demonstration of ANFs/TEOS-8, (**d**) Young’s modulus of ANFs aerogel and aramid fibers@silica aerogels obtained by different CVD times.

**Figure 6 polymers-15-00141-f006:**
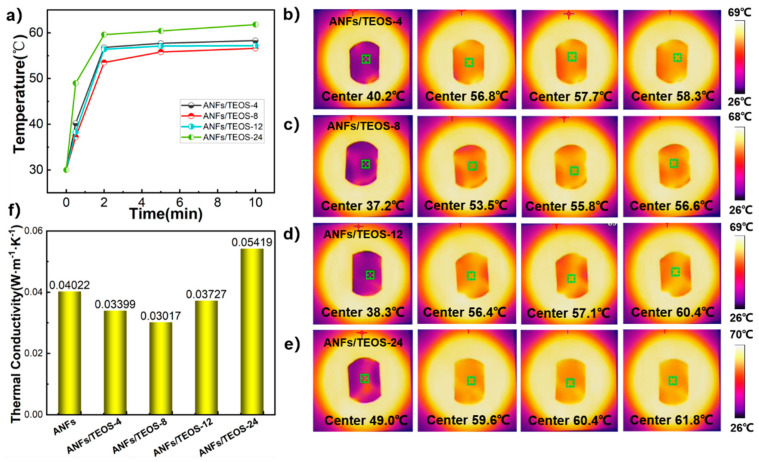
Thermal conductivity of the aerogels. (**a**) Curves of temperature change of one side of the samples with heating time of the other side. (**b**–**e**) Infrared thermal images of different aerogels. (**f**) Thermal conductivity of different aerogels.

**Figure 7 polymers-15-00141-f007:**
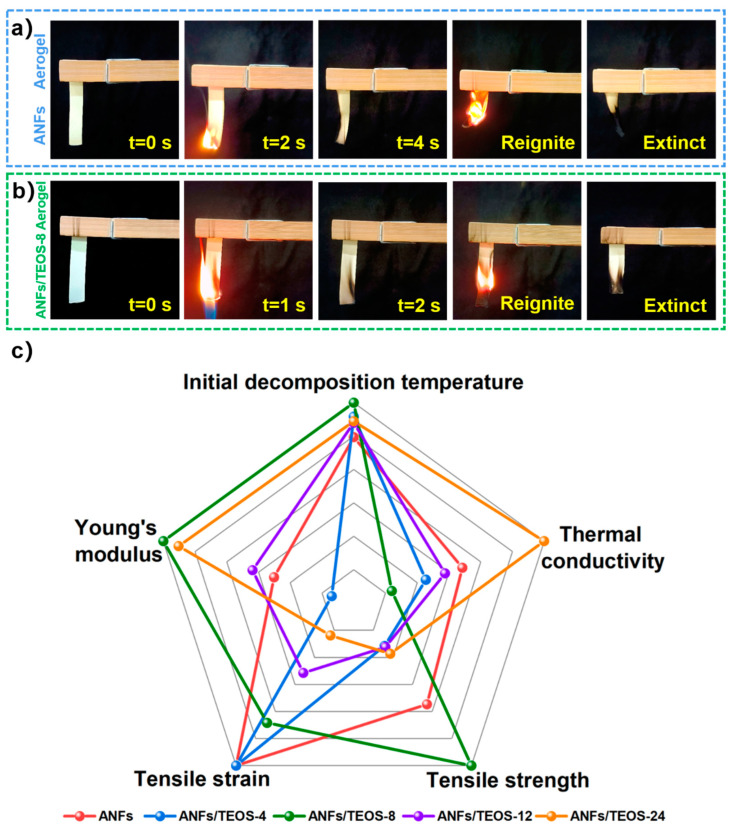
Flame-retardant experiments of (**a**) ANFs and (**b**) ANFs/TEOS-8 aerogel samples. (**c**) Radar map comparing the comprehensive properties of various aerogels.

**Figure 8 polymers-15-00141-f008:**
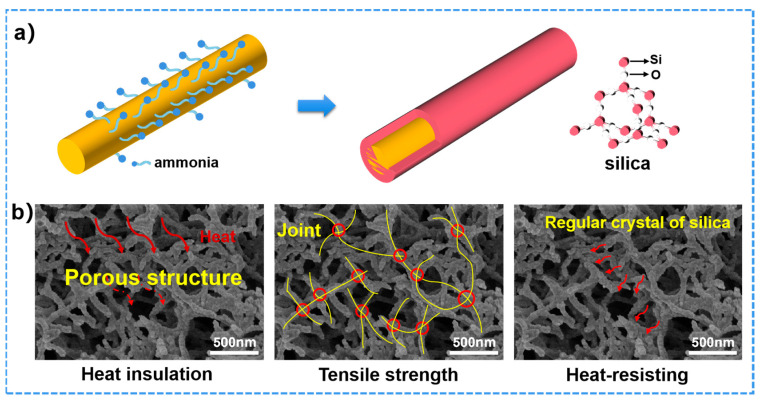
(**a**) Changes of small molecules on the surface of aramid fiber and shell-core structure of ANFs/TEOS during gas phase polymerization. (**b**) Structural schematic of the advantages of ANFs/TEOS-8.

**Table 1 polymers-15-00141-t001:** Density, initial decomposition temperature, thermal conductivity, tensile strength, and Young’s modulus of ANFs and ANFs/TEOS aerogels.

	Density (mg/cm^3^)	Initial Decomposition Temperature (°C)	Thermal Conductivity (W·m^−1^K^−1^)	Tensile Strength (MPa)	Tensile Strain (%)	Young’s Modulus (MPa)
ANFs	74.00	419.20	0.04022	5.80	11.7	49.57
ANFs/TEOS-4	96.92	448.01	0.03399	4.18	11.7	35.73
ANFs/TEOS-8	153.39	467.44	0.03017	7.50	9.9	76.14
ANFs/TEOS-12	216.86	439.31	0.03727	4.21	7.7	54.75
ANFs/TEOS-24	298.82	441.43	0.05419	4.40	6.1	72.49

## Data Availability

The data presented in this study are available on request from the corresponding author.

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
