# Peer review of "Bio-Inspired Aramid Fibers@silica Binary Synergistic Aerogels with High Thermal Insulation and Fire-Retardant Performance"

_polymers, 2022, doi:10.3390/polym15010141_

Round 1

Reviewer 1 Report

1. Line 74. It is necessary to specify the exact "room temperature"

2. Line 84.85. It is necessary to provide a calculated estimate of the error in determining the average thickness of the sample. 

3. Line 119 "However, the ANFs could not remain stable mechanical property in high temperature". It is not clear on the basis of which studies this property is justified in this article? The authors need to provide data at this point of the article confirming such properties for this particular sample of the obtained material. 

4. Section 3.4. It is necessary to describe the method, equipment and error estimation for determining the thermal conductivity of the fibers of the presented thickness. 

5. The list of references represents only 13.5% of the analysis of scientific research and results in the field of obtaining and studying the properties of aramid and aerogel fibers from different scientific schools of the world. This shows insufficient consideration of world achievements on the topic of the article. The list of references and consideration of a wider range of case studies in the article need to be increased and refined.

Reviewer 2 Report

The manuscript "Bio-inspired Aramid fibers@Silica Binary Synergistic Aerogels with High Thermal Insulation and Fire-retardant Performance" has an important and actual subject of the research field.

The general presentation of the work is good.

Methodologies and procedures were adequate.

The result are interesting and useful.

The reference covering the specific research field.

The conclusion were based on experimental results.

However some minor corrections are welcome:

- every used apparatus need the performance and producers (a little details).

- the "Results and discussions" section must a little rearrangement and expansion, because it starts too steeply.

- at introduction the references must be more specific (without interval, ex. [1-4]).

Reviewer 3 Report

Comments

In this paper, a novel bioinspired ANFs/TEOS aerogels film has been obtained by CVD of TEOS catalyzed by ammonia. The goal of this work is distinct and the logic is clear. However, there are still some issues to be addressed. The specific comments can be found as following:

1.       In order to rich the Introduction, some important references should be cited. Such as, (1) Top-down” fabrication of anisotropic, lightweight, super-amphiphobic, and thermal insulating rattan aerogels; (2) Water molecule-induced hydrogen bonding between cellulose nanofibers toward highly strong and tough materials from wood aerogel.

2.       The format of unit "Wm-1K-1" in Table 1 should be consistent with the preceding text and changed to " W·m-1·K-1".

3.       The scale in Figure 8b should be unified to facilitate comparative observation. The scale of “Heat insulation” can be changed to 500nm.

4.       How does directional laminar improve Young's modulus in line 186~188?

5.       In line 268, why is ANFs/TEOS-8 carbonized in argon atmosphere? Can other gases be used?

6.       In the infrared analysis, the author only analyzed the samples treated with 8-hour CVD. The author needs to further explain the reasons for selecting this sample, or the author can put the infrared analysis behind other analyses

7.       “For the aramid fibers@silica aerogels, the weight loss before 200℃is attributed to the evaporation of water in the 3D porous fiber. The maximum mass loss rate is found at around 520℃.” What is the reason for the fast decomposition rate around 520℃?

8.       “Moreover, the presence of shell-core structure and fixation of the nanofiber joint can hinder crack propagation to some extent, and Young's modulus is further promoted. ”The author needs to add non-putamen structure for comparison to highlight the role of putamen structure

9.       “and have certain electrical conductivity due to the interior carbon fiber.”The electrical conductivity test needs data support.

10.   It should be noted that the manuscript needs to be carefully edited by someone with expertise in technical editing in English, with special attention to English grammar, spelling, and sentence structure.

Round 2

Reviewer 1 Report

The article can be published

Reviewer 3 Report

Can be published after minor revision.